# Empathy and Mobile Phone Dependence in Nursing: A Cross-Sectional Study in a Public Hospital of the Island of Crete, Greece

**DOI:** 10.3390/healthcare9080975

**Published:** 2021-07-31

**Authors:** Michael Rovithis, Sofia Koukouli, Aristidis Fouskis, Ioulia Giannakaki, Kleanthi Giakoumaki, Manolis Linardakis, Maria Moudatsou, Areti Stavropoulou

**Affiliations:** 1Department of Nursing, School of Health Sciences, Hellenic Mediterranean University, Gianni Kornarou, Estavromenos 1, 714 10 Heraklion, Greece; rovithis@hmu.gr (M.R.); afouskis@yahoo.gr (A.F.); juliagian@hotmail.com (I.G.); kleiogiakoumaki10@gmail.com (K.G.); 2Department of Social Work, School of Health Sciences, Hellenic Mediterranean University, Gianni Kornarou, Estavromenos 1, 714 10 Heraklion, Greece; moudatsoum@hmu.gr; 3Department of Social Medicine, Faculty of Medicine, University of Crete, Andrea Kalokerinou 13, Giofirakia, 715 00 Heraklion, Greece; linman@med.uoc.gr; 4Department of Nursing, School of Health and Care Sciences, University of West Attica, Ag. Spyridonos Str., 122 43 Athens, Greece; astavropoulou@uniwa.gr

**Keywords:** empathy, mobile phones, altruism, dependence, nurses, healthcare

## Abstract

This study examined the relationship between empathy and mobile phone dependence levels of the nursing staff in a public hospital in the island of Crete, using a cross-sectional study design. Data from 109 staff nurses and healthcare assistants (HCAs) were collected via the Greek version of the Mobile Phone Dependence Questionnaire (MPDQ) and the Toronto Empathy Questionnaire (TEQ). Multiple linear regression was used to determine the correlation between empathy and mobile phone dependency. The total mean score for TEQ was 33.9 (±5.7). Accordingly, the total mean score for MPDQ was 22.9 (±6.1). High mobile phone dependence was found in 4.7% of the participants. A statistically significant difference was found between HCAs and staff nurses, with HCAs presenting a higher mean empathy levels (TEQ) (36.5 vs. 32.6) and lower dependence levels (MPDQ) (18.9 vs. 24.5) than staff nurses. A significant correlation between empathy and dependence was found between dependence and the altruism empathy subscale, with higher dependence being correlated with lower altruism. The participants’ levels of empathy do not seem to be affected by mobile phone dependence. However, empathy appears to be strongly determined by increased age and professional status. Nurses’ dependence on mobile phones is a complex phenomenon that requires attention. Educational programs on empathy and information on the proper use of mobile phones by the nursing staff should be provided.

## 1. Introduction

In a contemporary healthcare setting a mobile phone can be a useful tool due to its portability and immediate Internet access. Mobile phone applications allow healthcare professionals to monitor patients with chronic and life-threatening diseases and to prevent high-risk incidents. Such applications, for instance, may be used for patients with chronic heart failure and help nurses to identify early signs of arrhythmia or ischemia which may indicate an impending heart attack. Moreover, patients themselves can use mobile phone applications as well as monitoring and measuring devices to check their physical activities and physiological indices relevant to their condition [1].

In recent years, mobile phones applications have been used by researchers and professionals, as tools for introducing and supporting quality interventions in various healthcare settings. Healthcare professionals use mobile phones to provide education on diabetes, to help patients keep up with their treatment and follow-up appointments, and/or to provide guidance on healthy living and patient’s empowerment [2].

Despite the positive aspects and the benefits that mobile phones can provide through their functions and applications, several studies have documented various problems arising from their extensive use. A study on Internet usage showed that going online during working hours is becoming increasingly common, and most employees, irrespective of age or professional status, stated that they use their personal mobile phones or other personal communication devices for non-work-related activities in the workplace [3]. Mobile phones and other personal communication devices can distract healthcare providers and reduce the level of vigilance necessary for patient care. In a study on whether nursing students are distracted by mobile phone use, the researchers argued that these devices can be a significant source of interference during the performance of nursing tasks that require cognitive ability. This may result to reduced concentration, negative time reaction, and delays in task accomplishment [1]. In another study with a sample of 1268 nurses from two key nursing organizations in the United States, the majority of the respondents believed that mobile phones have more negative than positive impact on patient care and that they constitute “always” or “often” a source of distraction while working. The respondents acknowledged that the usage of their own device affected negatively their work performance, in terms of missing important clinical information, or making a nursing error [4].

Moreover, mobile phone use may lead to miscommunication between the “sender” and the “recipient” as face-to-face interaction becomes problematic [5]. When a person, during the communication process, is concentrated on his/her phone screen, this is interpreted by the other person, the “recipient” in the following ways: (a) the phone screen is more interesting than the current interaction; (b) he/she must wait a while; and (c) the phone is part of the interaction (for example when looking for information relevant to the conversation).

In the nurse–patient relationship, communication and empathy play an essential role in the establishment of the caring relationship, as well as for the provision of quality, holistic, and personalized nursing care [6].

Empathy is considered an essential skill and characteristic of health professionals. Its incorporation in daily clinical practice allows health professionals to recognize the effectiveness and significance of each intervention from the patient’s point of view [7]. However, empathy is hard to define accurately as researchers focus on and discuss its multidimensional nature involving cognitive, affective, and behavioral facets [8]. In one classic definition, empathy is a person’s ability to feel what another person is feeling, to sense the inner world of private personal meanings as if it were your own, but without ever losing the “as if” quality [9]. This definition highlights the emotional dimension of empathy without overlooking its cognitive and rational use, which includes the evaluation, monitoring, and analysis of interventions. Moreover, it incorporates the social dimension of empathy through the interaction between two persons, the one who displays empathy and the one who receives it [9].

To provide high quality of care in the clinical environment contemporary nurses seek to combine the rapidly changing technology with the need to reinforce empathy and communication with the patient through the development of strong interpersonal relations [10]. Many studies aim at investigating and assessing the levels of empathy in the nursing staff [10,11,12]. These studies examine important components of empathy, such as self-knowledge, understanding/perception, active listening, eye contact, and the time allocated to the patient. Nakamura highlighted that mobile phone use impedes face-to face communication, active listening, and eye contact between the nurse and the patient [5]. He further questioned the emotional and mental connection with the patient while the nurse is preoccupied with the mobile phone.

Research evidence on nurses’ empathy and mobile phone usage is rather limited in Greece. Relevant studies focus mainly on the exploration of the concept of empathy and its relative features within the field of nursing and social care. Empathy and implementation of empathic care in demanding clinical settings, such as the ICUs, was examined through nurses’ perceptions [7], while Greek nurse students’ views on empathy were also explored [13]. Furthermore, the notions of empathy in health and social care within the Greek context were conceptually analyzed and reviewed [14,15], and empathy and its variations were assessed in Greek social workers [16]. Evidence regarding the association of mobile applications with empathy and health outcomes in Greece is scarce [17], while research on empathy and mobile phone dependence in nursing is lacking. The present study aims to examine the relationship between empathy and mobile phone dependence levels of the nursing staff in a public hospital in the island of Crete, Greece. To the best of our knowledge, it was the first time that a study on nurses’ empathy and mobile phone dependence has been conducted in Greece. The results of this study will enrich the literature on empathy in nursing care and mobile phone usage, a subject which is under investigation within the Greek healthcare context.

## 2. Materials and Methods

### 2.1. Study Design, Sample, and Participants

This study was carried out in the island of Crete, in Greece, using a cross-sectional study design. The island of Crete, has five general tertiary hospitals, operating under the regulations of the Greek National Health System (NHS). The study was carried out at one of these hospitals which provides secondary and tertiary healthcare services, having a capacity of approximately 700 beds. This specific hospital was selected because it is the largest one in the island and is a reference (regional) hospital for the entire population of Crete.

Based on a convenience sampling strategy, a sample of 109 nurses was recruited, consisting of staff nurses and HCAs with at least one year’s experience in their professional position. HCAs are graduates from occupational training schools (public or private) with one or two years of basic education and hospital-based training prior to their employment. HCAs have an auxiliary role, providing patient care under the supervision of staff nurses. They are mainly involved in basic caring tasks such as sanitary care, patients’ feeding, patients’ dressing and grooming, and bed preparation. Staff nurses are graduates from either Universities or Technological Educational Institutes (recently reformed to Universities). The basic nursing course, in both institutes, is completed in a four-year study program, in which the 8th semester is dedicated to clinical practice. Staff nurses’ responsibilities focus on identifying nursing problems and formulating nursing diagnoses. Designing, implementing, and evaluating appropriate caring interventions for patients and monitoring patients’ condition, assessing patients’ needs, providing patient education and counselling are some of staff nurses’ duties.

The sampling framework and the field of study consisted of six clinics. The clinics that were included in the study represent the entire spectrum of adult patients’ medical and surgical nursing care (medical and cardiology clinics, surgical, maxillofacial and orthopedic clinics, intensive care unit, and psychiatric clinic).

### 2.2. Ethical Considerations

Before the commencement of the study ethical approval was granted (Ref number 2653/13-12-2016). Issues of anonymity and confidentiality were preserved throughout the research process before data collection. Additionally, the participants were also informed about the voluntary nature of the research and their right to withdraw from the study at any time without any consequence. Since the study involved human participants, all procedures performed were in accordance with the 1964 Helsinki declaration and its later amendments.

### 2.3. Data Collection

Data were collected by the researchers using two anonymous self-report questionnaires. The level of mobile phone dependence was measured using a validated tool, the Greek version of the Mobile Phone Dependence Questionnaire (MPDQ) [18]. This is a self-administered questionnaire consisting of 20 Likert-type questions on a scale of 0–3 with a Cronbach’s α of 0.89. The overall score is the sum of the answers and therefore ranges from 0–60. The higher the score, the higher the mobile phone dependence and vice versa [19]. Empathy was measured using the Toronto Empathy Questionnaire (TEQ) [20], consisting of 16 questions on empathy on a five-point scale from “never” to “always” and with a Cronbach’s α of 0.87 [21]. Permission to use the instruments was obtained via e-mail [18,21].

Before the commencement of data collection phase, several informal meetings were conducted by the research team with the nurse managers and the nursing staff in the selected study sites, for informing the potential participants about the aim and the nature of the study. During these meetings the significance of the research was explained and issues of voluntary participation, anonymity, and confidentiality were emphasized. Potential participants were also informed that hard copies of the study questionnaires would be available at a specific place in the nursing station of each clinic. Once completed, the questionnaires could be returned in a sealed envelope, in collection boxes, which were placed at accessible points of the clinics. At this phase of the study, many members of the nursing staff who attended the meetings, orally expressed their consent to participate in the research. Moreover, during the data collection phase, several reminders were used by the nurse managers and the researchers in order to obtain a satisfactory response rate. Τhe above actions as well as the existing academic collaboration among the hospital and the university led to the achievement of a high response rate.

One hundred twenty (120) anonymous questionnaires were distributed to nurses, while one hundred nine (109) were returned completed, resulting in a satisfactory response rate (90.8%).

The answers were completely confidential and only members of the research team had access to them. The collected data were used exclusively for scientific purposes. Publication of findings did not reveal any information that might hinder the anonymity of the study participants. The participants gave their informed consent by completing a questionnaire that included a cover page providing substantial information about the study and a statement to be filled out and returned. Protection of any personal data, confidentiality, and anonymity was preserved throughout the research process. The study data were collected between December 2017 and February 2018.

### 2.4. Data Analysis

Data were analyzed using SPSS software (IBM SPSS Statistics for Windows, Version 25.0. Armonk, NY, USA: IBM Corp.). Frequency distributions of the basic characteristics of the 109 participants were calculated, using chi-square (χ^2^) method, Mann–Whitney, and Kruskal–Wallis tests to detect any differences between genders or other characteristics. Distribution and classification were determined for each scale. Possible correlations between the scales were identified using Spearman’s rho. Multiple linear regression analysis was used to determine the correlation between empathy and mobile phone dependency.

## 3. Results

### 3.1. Demographic Characteristics of the Participants

Of the 109 study participants 84.4% were female. The mean age of all was 41.3 ± 7.5 (Table 1). Regarding marital status, 61.8% were married. Of the total participants 67.6% had children, while most respondents (94.1%) lived in an urban area. Professionally, 33.7% worked as HCAs and 66.3% were staff nurses. A total of 26.7% had worked in the same position between 6 and 7 years (Table 1).

### 3.2. Score Levels and Reliability of Empathy and Dependence Scales in Study Participants

The total mean score for TEQ was 33.9 (±5.7) and the total mean score for MPDQ was 22.9 (±6.1). The highest mean score was 11.7 (± 2.6) in Sympathetic physiological arousal subscale, followed by Altruism (7.7 ± 1.5). In the MPDQ, only 4.7% scored over 45. Scores for the total sample ranged from 16.8 to 29. Cronbach alphas reliability coefficient for the Greek edition of TEQ was 0.600 and for the Greek edition MPDQ was 0.853 (Table 2). 

### 3.3. Correlation of Participants’ Demographic Characteristics with Empathy and Mobile Phone Dependence

Correlating the participants’ demographic characteristics with their empathy and dependence scores, no statistically significant difference was found in age. However, there was a statistically significant difference (*p*-value = 0.012) in marital status, with unmarried respondents reporting higher mobile phone dependence, with a mean score of 27.9, than married respondents, with a mean score of 19.7. Respondents without children also scored higher on the MPDQ scale than those who had children, with mean scores of 26.8 and 20.3 respectively (*p*-value = 0.011).

A statistically significant difference was also found in both empathy (*p*-value = 0.002) and dependence (*p*-value = 0.032) based on profession: HCAs had higher empathy levels than staff nurses, with TEQ scores of 36.5 and 32.6, respectively. The MPDQ scale showed that staff nurses had higher mobile phone dependence, with a mean score of 24.5 compared to 18.9 for HCAs. No difference was found in empathy levels based on years of employment; on the contrary, however, there was a statistically significant difference (*p*-value = 0.034) for dependency, with the lowest score of 15.6 recorded in the 16–25-year group and the highest of 25.9 in the 6–10-year group (Table 3).

A significant correlation between empathy and dependence is only found between dependence and the Altruism empathy subscale, with higher dependence being correlated with lower altruism (r = −0.201, *p* = 0.038, results not shown on tables/figures). Multiple linear regression analysis of the empathy scale in study participants showed that their empathy does not seem to be affected by mobile phone dependence (b = 0.062, *p* > 0.05). However, empathy does appear to be strongly determined by increased age (b = 0.270, *p* = 0.046) and HCAs (professional status) (b = 0.307, *p* = 0.002) (Table 4).

## 4. Discussion

The aim of the present study was to examine the relationship between empathy and mobile phone dependence levels of the nursing staff, within the Greek healthcare setting. In summary, the following results were documented: (i) The total mean score on empathy was higher than the baseline; (ii) the level of mobile phone dependence among nursing staff appeared to be low with HCAs to demonstrate lower levels compared to staff nurses; (iii) higher empathy levels were documented in HCAs compared to staff nurses; (iv) higher empathy levels and lower levels of mobile phone dependence were documented in older age groups; (v) unmarried and divorced respondents demonstrated higher levels of mobile phone dependence than married participants; (vi) higher levels of mobile phone dependence appeared to be correlated with lower altruism levels.

### 4.1. Empathy

With regard to empathy, the findings showed that the total mean score on the TEQ scale was higher than the baseline (>15). Relevant studies which have been conducted at an international level, demonstrate that nurses’ empathy scores varied from high to low [10,22]. This variation is due to the difficulty in defining the term of empathy precisely and researchers may use various methods which often measure different aspects of empathy [23,24]. High levels of empathy were documented in a study conducted with a sample of 106 undergraduate nursing students at Monash University, Australia [25], while in Greece nurse students displayed a moderate level of empathy, with the final year students displaying more empathetic ability compared to the first-year students [26]. However, results for a longitudinal study which included 214 undergraduate nursing students at the Jefferson School of Nursing [27] showed a significant decline in mean empathy scores for undergraduate nursing students who were exposed more than others to patient encounters during study period despite the fact that more encounters with patients are supposed to strengthen empathic engagement.

Although there was no statistically significant difference depending on respondents’ marital status in the present study, higher empathy levels were observed in married and divorced individuals compared to unmarried ones. These findings are in accordance with a study [28], where married and/or divorced nurses appeared to have a higher mean empathy score on the Jefferson scale [29] than unmarried nurses. The researchers suggest this may be due to the experience of child-rearing. For example, teaching children to gain independence and undertaking increased family and societal responsibilities involve skills such as active listening and careful communication to discern the child’s needs. Parents therefore develop higher levels of empathy. Despite that, a similar study in operating room nurses in Turkey demonstrated contradictory evidence, by highlighting no correlation between marital status and levels of empathy [30].

In addition, a statistically significant difference was found between age and empathy levels, with the older participants being more empathetic than the younger ones. Similar studies however, revealed no correlation between age and empathy levels [31,32,33].

Another important finding is that HCAs demonstrated higher empathy levels compared to staff nurses. This might be related to the educational level, as relevant research [30], stressed that when the level of education increases, cognitive and emotional levels of empathy decrease. It should also be noticed that in the present study, the in-hospital work system involves the performance of many tasks by HCAs demanding more frequent and intense communication with patients. For instance, HCAs are responsible for the patients’ daily hygiene and/or measurement of vital signs, tasks that bring them closer to the patients, in contrast to staff nurses who are more focused on administrative duties and treatment tasks.

### 4.2. Mobile Phone Dependence

Regarding the level of mobile phone dependence, the results of the present study showed that the mean dependence score was relatively low. On the contrary, in a study with a sample of 825 nurses in medical and surgical departments, 78.1% used their mobile phone during working hours for non-work-related activities: mainly sending emails and text messages, reading news, checking/posting on social network sites, and playing games [34]. In the same study, 69.5% of nurses reported that mobile phone use had a negative impact on patient care [34].

When exploring possible correlations between mobile phone dependence and the demographic characteristics of the sample, no statistically significant difference between age groups was observed in this study. However, there is an inversely proportional difference between the 20–30 and 31–40 age groups: levels of mobile phone dependence decrease in older age groups.

Similarly, in a study [35] on mobile phone addiction in adolescence, younger age is documented as the most important factor in mobile phone addiction. According to this study, mobile phones have certain attributes and characteristics that make them especially attractive to adolescents. They provide identity and prestige in relationships with their peers, which is often a cause of addiction in these age groups. This view may explain the higher mobile phone dependence rates in nursing students that reported in relevant studies in Greece, compared to the ones described in the present study [18,36].

An important finding of the present study is that there is a statistically significant correlation between mobile phone dependence and marital status: unmarried and divorced respondents were more dependent than married participants. A possible explanation could be that unmarried and divorced participants have fewer responsibilities, a more intense social life, more leisure time, and a broader network of friends. Finally, in the present study HCAs appeared to have a lower rate of mobile phone dependence than staff nurses.

### 4.3. Empathy–Mobile Phone Dependence Correlation

A significant correlation between empathy and mobile phone dependence was found between dependence and the Altruism empathy subscale, i.e., higher dependence levels are correlated with lower altruism levels (r = −0.201, *p* = 0.038). Excessive mobile phone use may affect mutuality, the quality of interpersonal communication, and empathy [37]. The rapid development of technology and the extensive use of mobile phones in our daily lives have revolutionized the way we communicate and expanded the boundaries of human creativity [17]. Despite the obvious benefits, these new forms of communication contributed at the same time to increasing the feelings of loneliness and isolation and a corresponding decline in empathy. As the use of technology increases, empathy decreases and many people feel disconnected and isolated, mainly because technology such as the use of mobile phones creates the illusion of companionship, while actually reducing real communication and connectivity between people. However, empathy is not only an essential feature of the human condition in general, but is considered to be one of the fundamental tools of health professionals in particular, playing a major role in the creation and effectiveness of the therapeutic relationship [7,10]. Ιt has been proven that the development of an empathic relationship between the nurse and the patient is a fundamental component of the nursing practice, as it affects patient’s empowerment and impacts positively on the therapeutic process [7,14,15]. It is noteworthy that, in recent years, the worrisome finding that these new forms of communication are one of the factors contributing to the reduction of empathy has led scientists to attempt to integrate the element of empathy in technology by creating applications which support or enhance user’s empathic skills [17].

Moreover, high demanding and stressful working environments where an imbalance exists between job demands and the resources needed to meet them, may lead to professional burnout, disillusionment, reduced altruism, and the loss of capacity to feel empathy toward patients [38,39]. Within this context, job engagement fades over time and the professional moves progressively from feelings of involvement, devotion, and desire of accomplishment to feelings of disillusionment and emotional exhaustion. To move away from burn-out and maintain or regain job engagement nurses adopt various self-care habits [40]. From that perspective, the use of mobile phone in the workplace—especially from younger nurses—may be a way of coping, venting, detachment, and self-care. However, it seems that overuse can have the opposite results and affect negatively the communication with the patient and health outcomes. The level of mobile phone addiction seems to affect nursing adverse events and nurses’ burnout [41]. Thus, it is important to teach nurses, especially the younger ones, who use more often this technology, how to acquire effective self-care skills in order to maintain job engagement and prevent burn-out and the reduction of empathy as a result. Deeper understanding and recognition of patients’ experiences, worries, and perspectives, contribute to the precise assessment of patient needs and reinforces cooperation among patients and nurses. This leads to designing effective therapeutic plans and tailor-made interventions improving health outcomes and the quality of patients’ life [10].

### 4.4. Limitations of the Study

A limited number of staff nurses and HCAs from one public hospital in a single geographical area of Greece participated in the study. In this respect, our results are not representative of the nursing staff of the entire country. A convenience sampling strategy was used and participants were not randomly selected. Since this sampling technique imposes limits on generalizability, the results of this study should be viewed under these limitation. In addition, findings of the present study correspond to a first approximation, emphasizing mainly on the relationship between empathy and mobile phone dependence in nursing staff. An extended research in this field involving stratified sampling strategy may enhance generalizability of results. Therefore, in order to provide enriched evidence in the field under investigation, further research is recommended involving a larger and randomly selected sample.

## 5. Conclusions

In the present study, the level of mobile phone dependence among the members of the nursing staff was low. The mean empathy score was higher than the baseline. Moreover, the empathy of nursing staff does not appear to be determined by mobile phone dependence, apart from the altruism dimension of empathy, where higher levels of dependence are correlated with lower altruism levels. It is important for nursing staff to be aware of the impact of mobile phone dependence and to be able to identify the possible negative consequences that this may have on the quality of nursing care. At the same time, hospitals should develop and implement policies on proper mobile phone use in the workplace. Moreover, the findings of this study emphasized the need for educational interventions to reinforce staff nurses’ empathy and foster communication with their patients. Assessing nurses’ educational needs on empathy and empathic care is crucial for applying meaningful care to patients. Continuing education programs or in-service training based on nurses’ needs on empathic care, communication, and personal interaction may benefit nursing practice by strengthening the nurses’ empathic skills and improving thus the provided patients’ care. Nurse directors and managers are the key-stakeholders in developing and implementing appropriate educational and research initiatives within the clinical context. Motivating the nursing personnel through training and research on empathy, may provide the opportunity to redesign the nursing care in a more sensitive way and to cultivate empathetic communication and behavior. Finally, further studies should be conducted, focusing on the effect of mobile phone use on specific nursing care parameters, such as communication, patient assessment, patient participation in care planning and patient empowerment.

## Figures and Tables

**Table 1 healthcare-09-00975-t001:** Descriptive characteristics of the study participants.

		*n*	%
**Gender**	Male	17	15.6
	Female	92	84.4
**Age groups**	20–30	5	4.6
	31–40	43	39.4
	41–50	48	44.1
	51–60	13	11.9
	mean ± s.d.	41.3 ± 7.5	
**Marital status**	Married	63	61.8
	Unmarried	29	28.4
	Divorced	10	9.8
**Children**	yes	69	67.6
	no	33	32.4
**Area of residence**	Urban	96	94.1
	Rural	6	5.9
**Profession**	HCA	34	33.7
	Staff Nurse	67	66.3
**Years of employment**	0–5	19	18.8
	6–10	27	26.7
	11–15	18	17.8
	16–25	19	18.8
	26+	18	17.9
**Postgraduate studies**	yes	7	6.9
	no	94	93.1
**Doctoral studies**	yes	0	-
	no	101	100.0
**Department**	ICU	43	39.4
	Psychiatric	7	6.4
	Medical	20	18.3
	Maxillofacial Surgery/Orthopedic	22	20.2
	Cardiology	12	11.1
	Surgical	2	4.6

**Table 2 healthcare-09-00975-t002:** Score levels and reliability of Empathy and Dependence scales in study participants.

Scales & Subscales	Mean	Stand. Dev.	Median	Min	Max	Cronbach α
Toronto Empathy Questionnaire	33.9	5.7	33.0	23	48	0.600
Score > 15 (as >25th percentile)	*n* = 109 or 100.0%					
Emotional Assessment	4.8	2.4	4.0	0	11	0.508
Altruism	7.7	1.5	8.0	5	12	0.699
Sympathetic physiological arousal	11.7	2.6	12.0	5	16	0.683
Emotional Contagion	3.3	1.3	3.0	1	7	0.759
Emotion Comprehension	3.0	0.9	3.0	0	4	^a^
Empathetic Response	2.8	1.0	3.0	0	4	^a^
Mobile Phone Dependence Questionnaire (MPDQ)	22.9	6.1	21.0	0	59	0.853
Score < 45 (<75th percentile)	*n* = 102 or 95.3%					
high or ≥45	*n* = 5 or 4.7%					

^a^ Sub-scales with one item.

**Table 3 healthcare-09-00975-t003:** Empathy and dependence scale scores in study participants according to demographic characteristics.

			Empathy	Mobile Phone Dependence
		*n*	Mean (Stand. Dev.)	
**Age groups**	20–30	5	30.0 (7.0)	30.2 (18.5)
	31–40	43	33.2 (5.6)	23.4 (10.9)
	41–50	48	34.7 (5.6)	21.6 (12.1)
	51–60	13	34.7 (7.0)	18.4 (10.1)
	*p*-value		0.212	0.365
**Marital status**	Married	63	34.3 (5.8)	19.7 (11.6)
	Unmarried	29	32.5 (5.0)	27.9 (11.5)
	Divorced	10	35.2 (3.6)	23.4 (10.4)
	*p*-value		0.262	**0.012**
**Children**	yes	69	34.7 (5.5)	20.3 (11.6)
	no	33	32.3 (5.1)	26.8 (11.5)
	*p*-value		0.058	**0.011**
**Area of residence**	Urban	96	33.8 (5.5)	22.3 (12.0)
	Rural	6	34.8 (5.3)	24.7 (11.0)
	*p*-value		0.727	0.504
**Profession**	HCA	34	36.5 (6.0)	18.9 (11.3)
	Staff Nurse	67	32.6 (4.7)	24.5 (11.8)
	*p*-value		**0.002**	**0.032**
**Years of employment**	0–5	19	32.2 (5.5)	23.6 (12.8)
	6–10	27	34.2 (5.4)	25.9 (12.1)
	11–15	18	33.9 (5.7)	24.4 (8.2)
	16–25	19	35.1 (6.2)	15.6 (9.7)
	26+	18	34.1 (4.6)	21.6 (13.7)
	*p*-value		0.727	**0.034**
**Postgraduate studies**	yes	7	29.9 (6.4)	26.3 (12.6)
	no	94	34.2 (5.3)	22.3 (11.9)
	*p*-value		**0.026**	0.243
**Department**	ICU	43	32.6 (3.7)	24.2 (11.7)
	Psychiatric	7	35.1 (7.2)	27.4 (21.5)
	Medical	20	32.0 (4.9)	20.3 (11.9)
	Maxillofacial Surgery/Orthopedic	22	36.6 (7.4)	22.2 (8.9)
	Cardiology	12	35.5 (7.0)	18.1 (9.1)
	Surgical	2	35.0 (4.4)	18.6 (10.5)
	*p*-value		0.123	0.502

Mann–Whitney and Kruskal–Wallis tests. Bold, Statistically significant.

**Table 4 healthcare-09-00975-t004:** Multiple linear regression analysis of empathy scale scores of study participants.

	Empathy Scale		
	Stand. Beta	t	*p*-Value
Gender	0.014	0.14	0.890
Age	**0.270**	2.02	**0.046**
Marital status	−0.005	−0.05	0.963
Children	−0.123	−1.02	0.312
Area of residence	0.064	0.66	0.513
Profession	**−0.307**	−3.11	**0.002**
Years of employment	−0.152	−1.15	0.254
Postgraduate studies	0.130	1.33	0.188
Mobile Phone Dependence Scale	0.062	0.60	0.550
R^2^ adjusted	0.125		

Scale scores are presented in Table 1. Bold, Statistically significant.

## Data Availability

The data presented in this study are available on request from the corresponding author. The data are not publicly available due to privacy reasons.

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
