# Peer review of "Empathy and Mobile Phone Dependence in Nursing: A Cross-Sectional Study in a Public Hospital of the Island of Crete, Greece"

_healthcare, 2021, doi:10.3390/healthcare9080975_

Round 1

Reviewer 1 Report

This is an interesting manuscript, with some potential to be considered as innovative, in relation to the growing need of using ITCs as an essential part of many occupations. Of course, it should have some key (and negative) implications on these professional, that are worth to address in empirical research.

The paper has, however, some gaps and shortcomings, that must be addressed by the authors:

- In line 15, please clarify it corresponds to a mean comparison procedure; it is really difficult to read.

- To the best of my knowledge, empathy, dependence and other issues potentially related to ITC (including mobile phone) use are also related to self-care habits of such professionals. However, authors did not really address this relevant issue in their background. Although the available literature on self-care habits and health outcomes among nurses is really scarce, some studies such as https://hdl.handle.net/1956/2646 and (DOI) 10.3390/ijerph17103586 may result helpful for authors to enrich their literature review in this regard.

- Although authors used the TEQ and the MPDQ and detail some hints on their very basic properties, more evidence on their reliability indexes (not the same as Alphas that are more useful to determine reliability, but more robust), i.e., CRIs of their sub scales must be provided, referring whether to other studies, or calculating it with the present data.

- The data analysis strategy should be, definitely, improved. Authors limited this section to enumerating the analyses they used, not to discuss and provide insights on their accuracy, meaning and utility for the study.

- Please consider removing the exact p-values from the abstract, both for the case of mean comparisons and the bivariate correlations. Instead, authors could just detail all results, stating that they were all statistically significant in the text, i.e., referring to significant correlations.

- The sample size of this study was considerably small, and although some statistical procedures could get not really affected (e.g, t-tests remain reliable), their generalizability, or the capacity of being extrapolated to other groups different to the study sample itself, i.e., their external validity are really questionable. This statement applies to quantitative analyses (in this regard more and better discussion of these limitations is needed), and also to qualitative interpretations, such as the ones made by the authors. For these reasons, the conclusions of the paper should be reshaped to the sample-related context, clarifying that these results corresponded to a first approximation, and (at best) more insights are needed to support them. A good way to do so is encouraging other researchers to perform complementary investigation, so that it can be added to what is stated at lines 307-308.

Best wishes.

Author Response

We would like to thank the reviewer for the constructive comments. Please find below our replies to each one of them.  

REVIEWER 1

This is an interesting manuscript, with some potential to be considered as innovative, in relation to the growing need of using ITCs as an essential part of many occupations. Of course, it should have some key (and negative) implications on these professional that are worth to address in empirical research.

The paper has, however, some gaps and shortcomings that must be addressed by the authors:

  1. In line 15, please clarify it corresponds to a mean comparison procedure; it is really difficult to read.

Reply: We have rephrased as follows:  The total mean score for TEQ was 33.9 (±5.7).  Accordingly, the total mean score for MPDQ was 22.9 (±6.1).  High mobile phone dependence was found to 4.7% of the participants. We do hope that this is clear now.

------------------------------------------------------------------------------------------------------

  1. To the best of my knowledge, empathy, dependence and other issues potentially related to ITC (including mobile phone) use are also related to self-care habits of such professionals. However, authors did not really address this relevant issue in their background. Although the available literature on self-care habits and health outcomes among nurses is really scarce, some studies such as https://hdl.handle.net/1956/2646 and (DOI) 10.3390/ijerph17103586 may result helpful for authors to enrich their literature review in this regard.

Reply:  According to the reviewer’s suggestion, we have added a paragraph in discussion to show the relationship between empathy, self-care habits and mobile phone dependency. We chose to place this information in the discussion section as it seemed to serve more efficiently our data analysis (please see section 4.3. Empathy – Mobile phone dependence correlation).

---------------------------------------------------------------------------------------------------

  1. Although authors used the TEQ and the MPDQ and detail some hints on their very basic properties, more evidence on their reliability indexes (not the same as Alphas that are more useful to determine reliability, but more robust), i.e., CRIs of their sub scales must be provided, referring whether to other studies, or calculating it with the present data.

Reply: According to the reviewer’s suggestion, a new table (table 2) was introduced in the results section. In this new table, the score levels and reliability of Empathy and Dependence scales of the study participants are presented in an analytical manner. The authoring team consider that Cronbach alpha determine the reliability of scales and subscales used in the present study adequately.

-------------------------------------------------------------------------------------------------------------

  1. The data analysis strategy should be, definitely, improved. Authors limited this section to enumerating the analyses they used, not to discuss and provide insights on their accuracy, meaning and utility for the study.

Reply: With all due respect to the reviewer, the analysis was attempted to be consistent with the purpose and the research question, considering the limitations of the study (e.g. sample size).  Consequently, table 5 focuses on identifying significant correlations in relation to the aim of the study.

--------------------------------------------------------------------------------------------------------------

  1. Please consider removing the exact p-values from the abstract, both for the case of mean comparisons and the bivariate correlations. Instead, authors could just detail all results, stating that they were all statistically significant in the text, i.e., referring to significant correlations.

Reply: The text was corrected as per suggestion.

-------------------------------------------------------------------------------------------------------

  1. The sample size of this study was considerably small, and although some statistical procedures could get not really affected (e.g, t-tests remain reliable), their generalizability, or the capacity of being extrapolated to other groups different to the study sample itself, i.e., their external validity are really questionable. This statement applies to quantitative analyses (in this regard more and better discussion of these limitations is needed), and also to qualitative interpretations, such as the ones made by the authors. For these reasons, the conclusions of the paper should be reshaped to the sample-related context, clarifying that these results corresponded to a first approximation, and (at best) more insights are needed to support them. A good way to do so is encouraging other researchers to perform complementary investigation, so that it can be added to what is stated at lines 307-308.

Reply: Relevant modifications and additions were made as per suggestion.

----------------------------------------------------------------------------------------------------

Reviewer 2 Report

I have two comments that may improve the quality of the paper:  1. A reference to the job description of staff nurses and HCA is recommended in order for the reader to get a clear picture of the working responsibilities of these two groups. 2.The authors might also refer more extensively to the benefits of education on empathy and how this may alte

Author Response

We would like to thank the reviewer for the constructive comments. Please find below our replies to each one of them.  

REVIEWER 2

I have two comments that may improve the quality of the paper: 

  1. A reference to the job description of staff nurses and HCA is recommended in order for the reader to get a clear picture of the working responsibilities of these two groups.

Reply: We have clarified  this in section 2.1. Study Design, Sample and Participants

-------------------------------------------------------------------------------------------------------

  1. The authors might also refer more extensively to the benefits of education on empathy and how this may alter

Reply: We have referred more extensively to the benefits of education on empathy at the end of section 4.3. Empathy – Mobile phone dependence correlation as well as in  the Conclusion section .

-----------------------------------------------

Reviewer 3 Report

Thank you for the opportunity to review a manuscript entitled "Empathy and mobile phone dependence in nursing: A cross- 2 sectional study in a public hospital of the island of Crete, Greece." The manuscript is a good read but there are changes that I suggest to the authors.

The abstract and the introduction captured the research area and process well.

Study design, measures, data collection and analysis are satisfactory.

Methods: The study population

I suggest that the authors should describe the differences between HCA and Staff Nurse. I have a sense that the job titles are used differently in different regions. I also see that there is an imbalance in the proportion of the two groups and this could influence the outcome of the phenomenon under investigation and can affect the results.

Authors need to explain why there is an imbalance between the HCA and Staff nurse proportions.

Results

The authors report all the statistics or values that are in the table and there is no need present data in that way, e.g. If there are 84.4% females, the reader know that the remainder is male, and the table is available to the reader to follow on the other statistics.

Table 2 

Table 2 presents the male female differences of the staff consisting of N=17 males and N=92 Females.

I suggest that the authors should stich to doing the comparison between HCA and Staff nurse or combine the N=109 sample size and analyze the empathy scores. The male/female proportion is not necessary, it is [skew and the fact that all the p-value are not statistically significant show that it is redundant. I suggest that the authors replace the table and revise the results section accordingly.

Line 170: There is a use of the term Family status where as in the table it is captured as Marital status. The authors should select one term and use it consistently.

Line 176-177 and line 191: The authors use the term Occupation and Profession interchangeably.   Select one term and use it consistently.

The authors should revise the manuscript according to the suggestions provided above. 

Once Table 2 is corrected, the authors should also refocus the results, discussions,  conclusion and the abstract accordingly.

Author Response

We would like to thank the reviewer for the constructive comments. Please find below our replies to each one of them.  

REVIEWER 3

Thank you for the opportunity to review a manuscript entitled "Empathy and mobile phone dependence in nursing: A cross- 2 sectional study in a public hospital of the island of Crete, Greece." The manuscript is a good read but there are changes that I suggest to the authors.

The abstract and the introduction captured the research area and process well.

Study design, measures, data collection and analysis are satisfactory.

Methods: The study population

  1. I suggest that the authors should describe the differences between HCA and Staff Nurse. I have a sense that the job titles are used differently in different regions. I also see that there is an imbalance in the proportion of the two groups and this could influence the outcome of the phenomenon under investigation and can affect the results.

Authors need to explain why there is an imbalance between the HCA and Staff    nurse proportions.

Reply: We have clarified the difference between HCA and Staff Nurse in section 2.1. Study Design, Sample and Participants.

Regarding the imbalance in the proportion of the two groups, the analogy which is appeared in the present study represents the real situation which  exists in the majority of the public hopsitals in Greek health care system, as the number of staff nurses employed in our NHS is higher that the number of HCAs. This proportion does not affect the outcome of the phenomenon under invetsigationa according to the statistc tests which were performed.

-------------------------------------------------------------------------------------------------------

Results

  1. The authors report all the statistics or values that are in the table and there is no need present data in that way, e.g. If there are 84.4% females, the reader know that the remainder is male, and the table is available to the reader to follow on the other statistics.

Reply:   We have corrected the above mentioned issues as per suggestion.

-------------------------------------------------------------------------------------------------------

Table 2 

  1. Table 2 presents the male female differences of the staff consisting of N=17 males and N=92 Females.

I suggest that the authors should stich to doing the comparison between HCA and Staff nurse or combine the N=109 sample size and analyze the empathy scores. The male/female proportion is not necessary, it is [skew and the fact that all the p-value are not statistically significant show that it is redundant. I suggest that the authors replace the table and revise the results section accordingly.

Reply: Thank you the reviewer for this precise suggestion. Due to the small number of male participants it was difficult methodologically to do a comparison between male and female in empathy scores. Thus we have omitted this table along with relevant sections in the discussion part. In the light of this shortcoming we have also omitted the gender comparison regarding the empathy scores. 

-------------------------------------------------------------------------------------------------------

  1. Line 170: There is a use of the term Family status where as in the table it is captured as Marital status. The authors should select one term and use it consistently.

Reply: We have corrected this throughout the text as per suggestion.

-------------------------------------------------------------------

  1. Line 176-177 and line 191: The authors use the term Occupation and Profession interchangeably.   Select one term and use it consistently.

Reply: We have corrected this throughout the text as per suggestion.

-------------------------------------------------------------------------------------------------

  1. The authors should revise the manuscript according to the suggestions provided above.  Once Table 2 is corrected, the authors should also refocus the results, discussions, conclusion and the abstract accordingly.

 Reply: We have corrected the paper (results, discussion, conclusion and abstract accordingly).

Round 2

Reviewer 1 Report

For this revision of the manuscript, authors addressed and fulfilled most of my previous comments, to my satisfaction. However, there are some major issues that were not well-covered in the revisions, or were (perhaps) poorly understood by the authors. Therefore, I will list them below:

- The comment 4 of the past review report ("The data analysis strategy should be, definitely, improved. Authors limited this section to enumerating the analyses they used, not to discuss and provide insights on their accuracy, meaning and utility for the study") did not intend to ask the authors to remake the analysis, nor to state it had been incorrectly performed, even though some psychometrical gaps of the study (that I won't argue about, as they are quite common) are evident.

Instead, what I am asking about is to improve the section "2.4. Data Analysis", that offers vague and scarce information on what statistical parameters were considered, why, and what were the advantages they entailed to the study. For example, Mann-Whitney tests have an asymptotic relative efficiency greater than t-tests if data mets the assumption of normality. However, authors say nothing about the basic parameters to be considered in these cases (e.g., was multivariate normality met, or not?), in order to have information on the coherence and accuracy of the tests chosen by them.

- Similarly, there is not much information allowing to discard that (especially working with a sample of 109 cases) there might be a failure on the independence assumption assessed by means of the X2 test, potentially explaining incidental correlations, that is always something to consider discarding, except for the case of randomized trials (that is not the case of this study), so that it should be cautiously assessed and detailed.

- Also, what were the statistical power calculations considered at sample size calculations (and how were them carried out)? What convenience sampling-related statistical remedies were used to minimize common method biases, and what was their background?

- Authors state in their response "Consequently, table 5 focuses on identifying significant correlations in relation to the aim of the study", but there is no Table 5 in the manuscript.

- Both MPDQ and TEQ reliability measures are based on Cronbach’s α that, although having considerably acceptable values in the present application, remain highly criticized as a single indicator for assessing scale reliability (see Raykov (2001); 10.1177/01466216010251005). Could for instance composite reliability indexes calculated?

- Discussion remains able to be improved, especially for what concerns the validation of the study findings in the light of previous studies cited (and not yet cited) in the manuscript.

Best wishes.

Author Response

Thank you for the very constructive comments. We believe they helped us improve the previous version. Please find below our replies to each one of them. 

REVIEWER 1

Comments and Suggestions for Authors

For this revision of the manuscript, authors addressed and fulfilled most of my previous comments, to my satisfaction. However, there are some major issues that were not well-covered in the revisions, or were (perhaps) poorly understood by the authors. Therefore, I will list them below:

COMMENT 1. The comment 4 of the past review report ("The data analysis strategy should be, definitely, improved. Authors limited this section to enumerating the analyses they used, not to discuss and provide insights on their accuracy, meaning and utility for the study") did not intend to ask the authors to remake the analysis, nor to state it had been incorrectly performed, even though some psychometrical gaps of the study (that I won't argue about, as they are quite common) are evident.

Instead, what I am asking about is to improve the section "2.4. Data Analysis", that offers vague and scarce information on what statistical parameters were considered, why, and what were the advantages they entailed to the study. For example, Mann-Whitney tests have an asymptotic relative efficiency greater than t-tests if data mets the assumption of normality. However, authors say nothing about the basic parameters to be considered in these cases (e.g., was multivariate normality met, or not?), in order to have information on the coherence and accuracy of the tests chosen by them.

Reply: With all due respect to the reviewer’s point of view, we believe that it is unnecessary to explain in details in the statistical analysis section (2.4), why our data suggested the use of non-parametric methods (e.g. Mann-Whitney test) instead of parametric ones (e.g t-tests). Since we use a non-parametric test it is implied that in our sample the assumption of normality was not met, and in that case non-parametric tests are more powerful than parametric tests.

COMMENT 2. Similarly, there is not much information allowing to discard that (especially working with a sample of 109 cases) there might be a failure on the independence assumption assessed by means of the X2 test, potentially explaining incidental correlations, that is always something to consider discarding, except for the case of randomized trials (that is not the case of this study), so that it should be cautiously assessed and detailed.

Reply: The reviewer is right that correlations statistics do not always indicate a cause-effect relationship between the variables. And, as with all inferential statistics, the results from a chi-square test are most reliable when the data are collected from randomly selected subjects, and when sample sizes are sufficiently large that they produce appropriate statistical power. As the above are well known we consider that it was not necessary to refer to the limitations of the method to demonstrate causality.

COMMENT 3. Also, what were the statistical power calculations considered at sample size calculations (and how were them carried out)? What convenience sampling-related statistical remedies were used to minimize common method biases, and what was their background?

Reply: As pointed out in the Methods section, this is a non-probability sample (convenience sampling method). Therefore, it is irrelevant here to refer to the “power of sampling”. The limitations of convenience sampling in the generalizability of results are well known and conclusions drawn from convenience sampling can be valid only for that sample (please see ‘limitations of the study’).

COMMENT 4. Authors state in their response "Consequently, table 5 focuses on identifying significant correlations in relation to the aim of the study", but there is no Table 5 in the manuscript.

Reply: The reviewer is right. Response is referred to table 3. 

COMMENT 5. Both MPDQ and TEQ reliability measures are based on Cronbach’s α that, although having considerably acceptable values in the present application, remain highly criticized as a single indicator for assessing scale reliability (see Raykov (2001); 10.1177/01466216010251005). Could for instance composite reliability indexes calculated?

Reply: Validity issues have been resolved by the creators of the specific scales used (MPDQ and TEQ). The scales were also validated in Greek. In addition, the small sample of participants in this study would not allow the use of the structural equation methodology. However, we consider that we have used two valid and reliable evaluation tools, as they are mentioned in the relevant literature.

COMMENT 6. Discussion remains able to be improved, especially for what concerns the validation of the study findings in the light of previous studies cited (and not yet cited) in the manuscript.

Reply: Please see the changes made in the Discussion section especially in 4.3.